# Hybrid Sharing for Multi-label Image Classification

**Zihao Yin**[2,3]**, Chen Gan**[2,3]**, Kelei He**[1,3]*, **Yang Gao**[2,3] and **Junfeng Zhang**[1,3]

[1] Medical School of Nanjing University
[2] State Key Laboratory for Novel Software Technology, Nanjing University
[3] National Institute of Healthcare Data Science, Nanjing University

```
{zihao.yin, chengan}@smail.nju.edu.cn
{hkl, gaoy, jfzhang}@nju.edu.cn
```

## Abstract

Existing multi-label classification methods have long suffered from label heterogeneity, where learning a label obscures another. By modeling multi-label classification as a multi-task problem, this issue can be regarded as a negative transfer, which indicates challenges to achieve simultaneously satisfied performance across multiple tasks. In this work, we propose the Hybrid Sharing Query (HSQ), a transformer-based model that introduces the mixture-of-experts architecture to image multi-label classification. HSQ is designed to leverage label correlations while mitigating heterogeneity effectively. To this end, HSQ is incorporated with a fusion expert framework that enables it to optimally combine the strengths of task-specialized experts with shared experts, ultimately enhancing multi-label classification performance across most labels. Extensive experiments are conducted on two benchmark datasets, with the results demonstrating that the proposed method achieves state-of-the-art performance and yields simultaneous improvements across most labels. The code is available at this URL.

## 1 Introduction

In computer vision, multi-label classification (MLC) attempts to predict multiple labels that may simultaneously appear in a single sample. It is more realistic and intuitive as a sample typically has multiple attributes in real scenarios. However, the semantic correlation and heterogeneity among different labels pose a significant challenge to MLC, resulting in the labels either complement or conflict with each other. Previous works (Liu et al., 2021a; Ridnik et al., 2023; Ye et al., 2020) achieved impressive performance via transformers or graph neural networks, trying to explore the correlation among labels with shared backbone across labels. These approaches neglected the heterogeneity among labels, which becomes the key obstacle to simultaneous improvement across labels.

In contrast to traditional multi-label classification approaches, MLC can be formulated as a multi-task learning (MTL) problem by modeling the prediction of each label as an individual task. The correlation and heterogeneity of the labels in MLC thus correspond to the task transfer problem of MTL, where learning a new task may perfect (positive transfer) or deteriorate (negative transfer) another. Under this context, the power of MTL in mitigating negative transfer may help improve the performance of MLC.

Precedent works like (Ma et al., 2018) in MTL include a mixture of experts (MoE, (Jacobs et al., 1991)), which utilizes a group of learned experts to handle different tasks separately. MOE has been widely adopted in natural language processing, where experts are expected to process words of various lexical categories. We advocate employing MoE in MLC image classification, which shares commonality with lexical category handling. Furthermore, we notice that the conventional MoE approach has primarily emphasized the utilization of expert groups within a specific task, with limited attention to the exchange of expertise group knowledge across different tasks. This approach may not align seamlessly with the MLC requirements, which will be scrutinized in our work.

---

*: Corresponding author

In this work, we introduce Hybrid Sharing Query (HSQ), a MoE-based MLC method with a novel proposed fusion strategy to better exploit semantic correlation and heterogeneity among labels and generate better underlying shared representation and task-specific representation. Additionally, we prioritize the adaptive fusion of label-specific and shared features in the classification task of each label, suppressing negative transfer and enhancing performance on the majority of labels. Specifically, we employ a group of shared experts to mine correlation among labels to generate multiple distinct shared features while assigning a group of task-specialized experts to each task to extract a series of label-specific features. This design can balance label-specific and shared features across labels while also emphasizing unique label-specific features for each individual label. Moreover, we employ gate networks to adaptively re-weight and harmonize features from task-specialized experts and shared experts, enhancing positive correlations and suppressing negatives among tasks.

Experiments show that the proposed method outperforms all tested baselines across multiple datasets on the majority of labels. The proposed method is also compatible with transformer-based MLC methods, indicating potential improvement to existing works.

Our contribution is three-fold:

• We present MoE to the MLC task, with gated task-specialized and shared experts to capture correlation and heterogeneity adaptively by formulating the MLC as an MTL problem.

• We empirically demonstrate that the fused experts help to extract correlations between tasks, encourage positive correlation sharing and suppress negative transfer, which benefits the overall and per-label performance and mitigates cross-label performance gap.

• We verify the superiority of our proposed model on two benchmark datasets with state-of-the-art performance overall and per-label.

## 2 RELATED WORK

**Multi-label classification in computer vision.** Models via various approaches have been proposed to address MLC. Zhu et al. (2017) use convolutional networks on an attention map to optimize ResNet prediction. Rajpurkar et al. (2017) solve the medical multi-label problem by using DenseNet (Huang et al., 2017). Wang et al. (2016) attempt to extract features from the image and generate the label as a sequence through a learned joint embedding space. Chen et al. (2019a) introduce graph convolutional network into this task, mapping label word embedding to inter-dependent object classifiers. Lanchantin et al. (2021a); Liu et al. (2021a) introduce transformer into MLC. These methods fail to see the negative transfer and positive correlation among labels. Some works also notice a similar problem in MLC from the MTL aspect. Wu et al. (2019) try to mitigate such a problem via a different architecture. Our study aims to improve overall performance in MLC while attempting to simultaneously enhance performance on as many labels as possible.

**Multi-task learning.** MLC can be recognized as a special case of MTL, treating each label as a separate classification task (Wu et al., 2019). Previous works on this topic include hard and soft parameter sharing, *etc*. Hard parameter sharing (Caruana, 1997) comes with a shared feature extraction backbone as a bottom and task-specialized towers as a top. Soft parameter sharing does not explicitly share network components across tasks but jointly learns other information through gradient sharing or other techniques. Duong et al. (2015); Yang and Hospedales (2017) encourage knowledge sharing across experts via different constraints like L2 norm. Cross-Stitch (Misra et al., 2016) trains two networks for two tasks and shares gradients between some layers controlled by gates. However, these architectures require more attention to the correlations among tasks, and the naive knowledge-sharing strategy may hamper the performance of models. In this work, we propose HSQ to reveal these correlations in the hope of generating a better representation for each task.

**Mixture of experts in deep learning.** Efforts have been made to improve models' performance by scaling up the model size with MoE (Jacobs et al., 1991), which first attempts to combine the outputs of several experts with a gate network. MMOE (Ma et al., 2018) with similar settings further decouples the seesaw phenomenon between several tasks by assigning exclusive gate and tower networks to each task. MOEC (Xie et al., 2023) adopts a clustering loss to impose variance-based constraints on the routing stage, obtaining clusters of experts with more diverse knowledge.

PLE (Tang et al., 2020) adds shared and task-specific experts to MMOE to allow better information sharing between tasks. Traditional MoE imposes a substantial computational burden since all experts activate, even when only some tasks are required. To mitigate such a cost, the sparse MoE (Shazeer et al., 2017) strategy emerges in contrast to the regular *dense* one. The routing strategy determines which experts contribute to the task output. Zhou et al. (2022); Rosenbaum et al. (2018); Nie et al. (2021); Zuo et al. (2022); Roller et al. (2021); Dai et al. (2022) and others explore various routing strategies, including randomizing, hashing, expert-choosing, *etc*. Switch Transformer (Fedus et al., 2022) introduces a sparse MoE to the transformer layer to replace the feed-forward neural network. Our method introduces the MoE into the multi-label classification field by virtue of task-specialized and shared experts exploiting correlations among tasks. Moreover, we utilize a gate network to enhance positive correlation and suppress negative correlation in pursuit of better fusion.

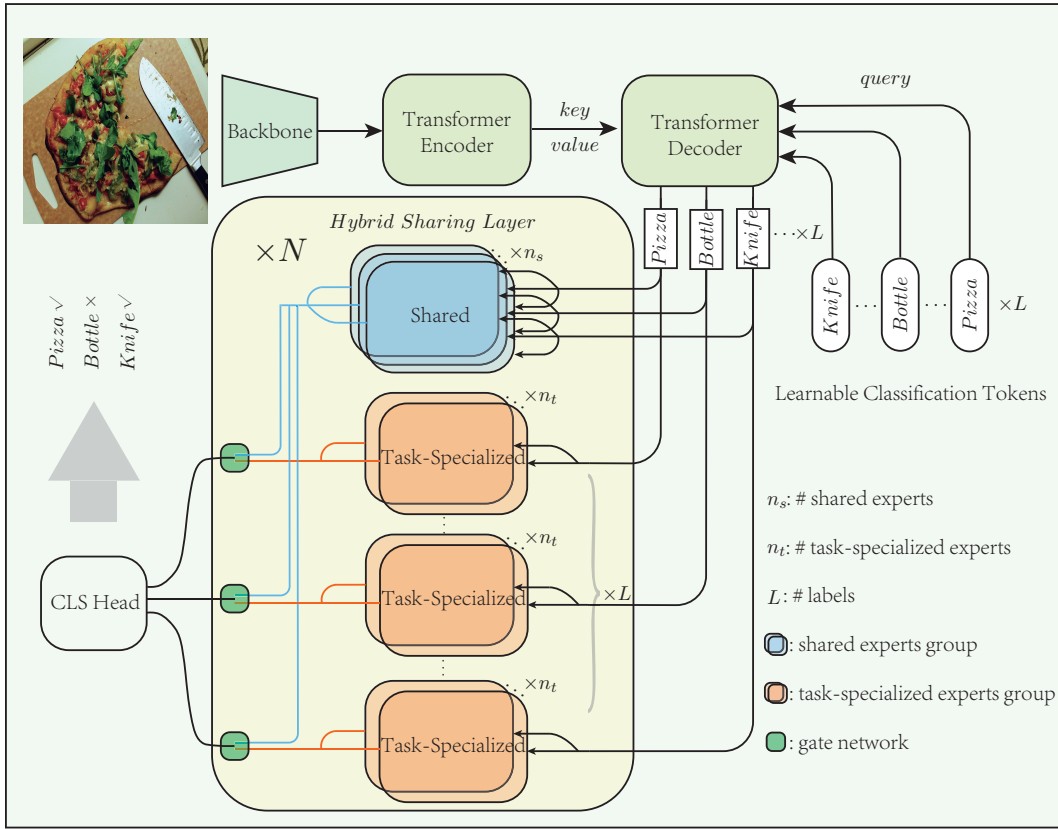

Figure 1: The architecture of the proposed HSQ. After being extracted by the backbone, the input image's features are then processed by a transformer, where learnable classification tokens are used as the query. $N$ Hybrid Sharing Layers are employed sequentially, consisting of $L$ groups of task-specialized experts and a group of shared ones. Individual gates for every group control the weighted outputs. A final classification head is utilized to make predictions.

## 3 METHOD

The MLC task for images is to find all possible correct labels in a pre-defined label set for a provided image. Thus, our model takes the provided image $I$ and gives probability scoring $\hat{\mathcal{L}} \in \mathbb{R}^L$ on all labels $\mathcal{L}$. The proposed HSQ model comprises three main parts, namely, 1) feature extraction backbone, 2) query transformer, and 3) mixture-of-expert hybrid sharing head. The backbone extracts image representation with a robust replaceable network, followed by a transformer-based query model to explore underlying information between such extracted representation and each given label. The Hybrid Sharing Layers are applied to better exploit the correlations between every possible task and suppress potential negative transfer problems.

## 3.1 Feature extraction backbone

Features in any given image will be extracted through a feature extraction backbone. Multiple preceding works have contributed to this stage. We employ various well-established models to capture global and local feature information within images more effectively. For a 3-channel input image $I \in \mathbb{R}^{3 \times H_i \times W_i}$, $H_i$ and $W_i$ are the height and width of an image. A feature extractor is applied to extract feature $R \in \mathbb{R}^{C_i \times H \times W}$, where $C_i$ denotes the number of feature embedding, with a succeeding convolutional layer linearly projecting its feature space from $C_i$ into $C$.

## 3.2 Query transformer

The semantic heterogeneity across labels requires the model to discern and capture unique feature representations specific to each individual task. Inspired by the remarkable performance of the query-based classifier, we employ learnable query tokens for classification to mitigate semantic conflicts between tasks. Specifically, this work employs a transformer to better extract and wrap task-specific underlying features in class-wise learnable tokens.

Given an extracted image representation $R$, an encoder-decoder standard transformer is applied to inspect features for each label. On the encoder side, the image representation from the backbone is flattened into $R \in \mathbb{R}^{C \times HW}$ and proceeded by $N_e$ encoder layers as tokens. To decouple different labels effectively, we endorse Liu et al. (2021a); Lanchantin et al. (2021b) to use learnable tokens as the query. On the decoder side, a learnable token is fed to the transformer decoder as the query for each possible label so that the feature of each label would be learned individually. $N_d$ decoder layers are stacked to extract the features of input representations in accordance with each possible label. The decoder layer accepts $T \in \mathbb{R}^{L \times C}$ for every $L$ possible label, where $C$ is the embedding dimension for each token. The cross-attention module in the transformer decoder performs on the query from the learnable label tokens (decoder) and the key and value from the extracted features (encoder), facilitating each label to mine respective representations.

## 3.3 Hybrid Sharing Layer

Given the potential semantic correlations among different labels, the features extracted from corresponding tasks may exhibit a positive correlation, providing complementary information to enhance model performance. However, improper exploitation of these correlations through learning jointly may cause performance degeneration since parts of these labels conflict with each other semantically due to their inherent heterogeneity, making them hard to learn jointly. To better leverage the positive correlations while suppressing detrimental impact due to heterogeneity between different tasks, we introduce the MoE mechanism into the multi-label classification area, inspired by the success of Progressive Layered Extraction (PLE) (Tang et al., 2020). Particularly, we employ several shared and task-specialized experts to capture positively correlated features among tasks and task-specific features, respectively, with a gate network adaptively fusing these features. The design of experts and gates can be very flexible and compatible as long as the output shapes are aligned, and in this work, we employ simple but effective linear layers to illustrate our approach.

Figure 2 depicts the details of Hybrid Sharing Layers, where $L$ indicates the number of tasks, *i.e.* the number of labels in multi-label classification. For any task $t_i, i \in \{1, 2, \cdots L\}$, a group of task-specialized experts $E_{t_i,j}, j \in \{1, 2, \cdots n_t\}$, is assigned to extract features for this task exclusively, where $n_t$ refers to the number of experts for this task. Apart from these task-specialized experts groups for every task, a group of shared experts $E_{s,j}, j \in \{1, 2, \cdots n_s\}$ is responsible for gathering global patterns and dispatching them to those potentially positively correlated tasks. Outputs of each expert group are harmonized by gate network, respectively, so that each task would have customized control on the weight of task-specialized and shared experts' outputs.

Algorithm 1 outlines the detailed mechanism of the MoE mechanism that we applied. Let $X_t \in \mathbb{R}^{N \times L \times d_i}$ be a batched input to the mixture-of-expert layer, where $N$ refers to batch size and $d_i$ means input embedding dimension. And let $X_s$ be the input for shared experts group with the exact same shape as $X_t$. The outputs of a Hybrid Sharing Layer comprise task-specialized outputs and a shared output.

In the task-specialized section, each label (task) is processed independently. For a batched input $X_{t_i} \in \mathbb{R}^{N \times d_i}$ on task $t_i$, a set of task-specific experts, denoted as $E_{t_i,j} \in \mathbb{R}^{d_i \times d_o}$, is utilized,

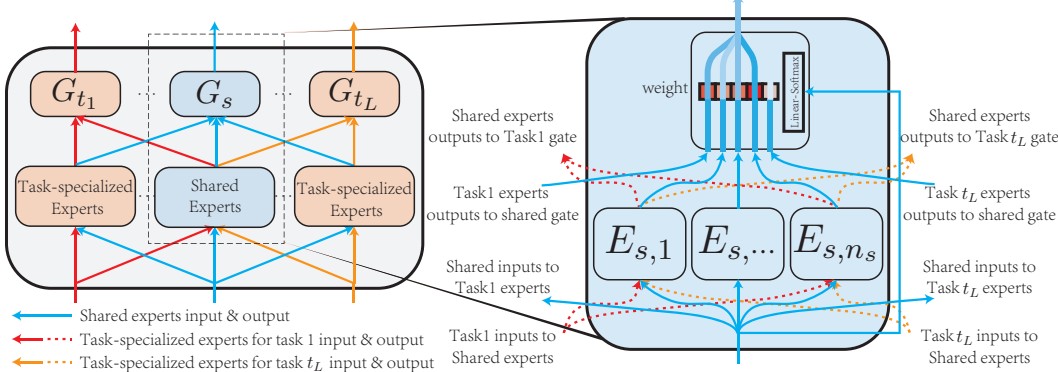

Figure 2: The Architecture of a Hybrid Sharing Layer. For $L$ labels, the layer consists of $L$ groups of task-specialized experts and a group of shared experts. The detailed structure of the shared experts is illustrated on the right.

where $j$ represents the $j$-th expert in the group. $Y_{t_i|t_i,j} = X_{t_i}E_{t_i,j} \in \mathbb{R}^{N \times d_o}$ represents the output of expert $E_{t_i,j}$ on task $t_i$. The subscript of $Y$, separated by $|$, refers to the output task and expert subscript, respectively, meaning that it is the output of the $j$-th task-specialized expert in $t_i$ and takes inputs from task $t_i$. Similarly, a group of shared experts, denoted as $E_{s,j} \in \mathbb{R}^{d_i \times d_o}$, is used for task $t_i$. Shared experts, which would be used in all tasks, also accept $X_{t_i}$ in task $t_i$, and $Y_{t_i|s,j} = X_{t_i}E_{s,j} \in \mathbb{R}^{N \times d_o}$ represents the output of expert $E_{s,j}$ on task $t_i$. A gate network, denoted as $G_{t_i} \in \mathbb{R}^{d_i \times (L \cdot n_t + n_s)}$, is employed to produce weights for outputs from all shared and task-specialized experts on task $t_i$. The gate network takes $X_{t_i}$ as input, and outputs $\text{Softmax}(X_{t_i}G_{t_i}) \in \mathbb{R}^{N \times (n_t + n_s)}$ as the weights for experts' outputs. Here, $n_t$ task-specialized experts for $t_i$ and all $n_s$ shared ones are employed. The task output is a weighted mean of all experts with activation $\sigma$, as described in the following equations, where $(k)$ stands for tensor indexing.

$$Y_{t_i} = \text{Concat}(Y_{t_i|t_i,j}, Y_{t_i|s,j}) \in \mathbb{R}^{N \times d_o \times (n_t + n_s)}$$
$$O_{t_i} = \sum_k \left[ \sigma(Y_{t_i})^{(k)} \odot \text{Softmax}(X_{t_i}G_{t_i})^{(k)} \right] \in \mathbb{R}^{N \times d_o} \qquad (1)$$

In the shared section, all shared experts $E_s$ and task-specialized experts $E_{t_i}$ are utilized to gather potential features, with a total of $n_s + L \times n_t$ experts. These experts use shared input $X_s$ as their input. Similar to the task-specialized part, a gate fuses shared and task-specialized features. The shared gate network, denoted as $G_s$, harmonizes the outputs from both shared experts and task-specialized experts across all tasks with weights derived from the shared input $X_s$. Algorithm 1 described shared and task-specialized parts in the Hybrid Sharing Layer.

$$Y_s = \text{Concat}(Y_{s|s_j}, Y_{s|t_{i,j}}) \in \mathbb{R}^{N \times d_0 \times (L \cdot n_t + n_s)}$$
$$O_s = \sum_k \left[ \sigma(Y_s)^{(k)} \odot \text{Softmax}(X_sG_S)^{(k)} \right] \in \mathbb{R}^{N \times d_0} \qquad (2)$$

It is worth noting that the shared and task-specialized parts receive the outputs from their respective parts in the previous layer as inputs, except the initial layer, which uses an identical input.

## 4 EXPERIMENT

We have performed extensive experiments on two datasets, MS-COCO and PASCAL VOC, to verify the superiority of our model. In accordance with the preceding works, we choose mean average precision as our primary metric. Some experiments also report some secondary metrics, including overall F1-score (OF1) and pre-category F1-score (CF1). Metrics on Top-3 are also reported. The definitions of these metrics are available in the Appendix.

**Algorithm 1:** Hybrid Sharing Layer Procedure

**Data:** Input to shared experts $X_s$; Input to task $t_i$ experts $X_{t_i}$; Shared expert gate $G_s$; Task $t_i$ expert gate $G_{t_i}$; Number of shared experts $n_s$; Number of task-specialized experts per task $n_t$; Shared experts $E_{s,(\cdot)}$; Task-specialized experts $E_{t_{(\cdot)},(\cdot)}$; Number of labels $L$; Activation function $\sigma$

**Result:** Shared output $O_s$; Task-specialized output $O_{t_i}$

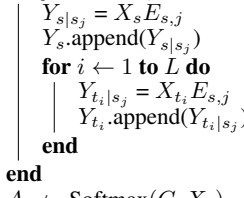

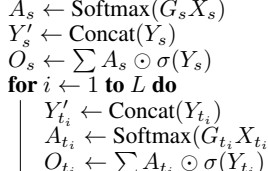

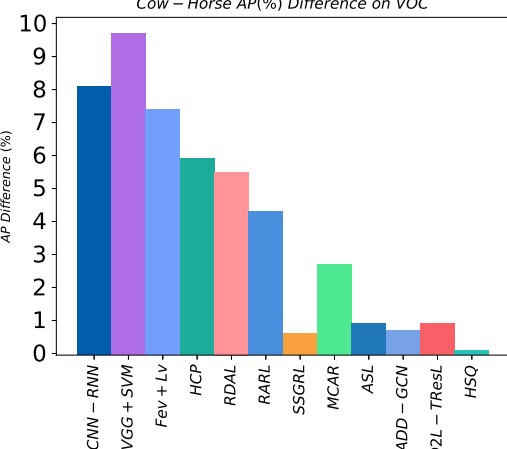

Figure 3: Absolute AP performance difference between cow and horse on VOC dataset in (%)

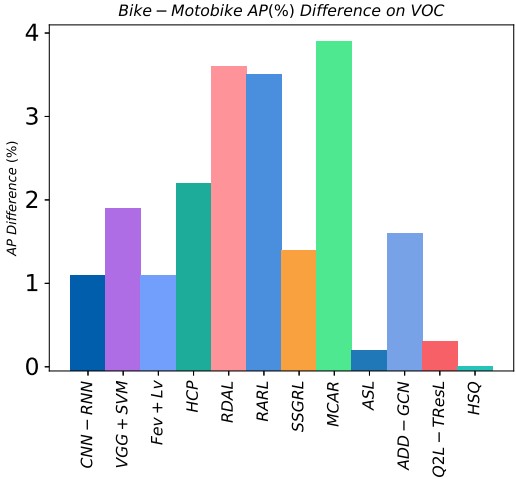

Figure 4: Absolute AP performance difference between bike and motobike on VOC dataset in (%)

## 4.1 ABLATION STUDY

As shown in Table 1, the number of shared experts greatly influences the performance. We choose ResNet10T as the backbone and Q2L with the same backbone as the baseline. An ablation study is performed on MS-COCO. The input image is fixed at a size of $576 \times 576$. The proposed model, which includes shared experts (HSQ), outperforms the baseline by 1.3% on mAP, demonstrating that including shared experts facilitates the transfer of information between tasks and mitigates negative transfer. The results also reveal that removing shared experts from the model leads to a considerable drop in performance due to the complete cutoff of sharing information among all tasks, underscoring the importance of sharing features in achieving substantial performance improvements. HSQ-Linear indicates that the hybrid sharing layers of the model are replaced by fully-connected layers with the same depth and dimensions, sharing all information across all labels without discriminating task-specialized information. It is demonstrated that the inclusion of shared experts is a crucial factor in enhancing the performance of the proposed model compared with HSQ-Linear. The findings highlight the potential benefits of incorporating shared experts and can inform the development of future multi-label image classification models.

## 4.2 PERFORMANCE ON THE MS-COCO DATASET

MS-COCO (Lin et al., 2014) is a large dataset of 80 object classes originally for image segmentation and object detection tasks. By extracting object information in annotations, it is also widely used to evaluate various models for multi-label image classification tasks. We test our model on MS-

Table 1: Ablation Study on MS-COCO. $n_s, n_t$ stand for the number of shared experts and task-specialized experts per task, †indicates that it is not available in the original work and we implement it in this paper.

| Method | Backbone | Resolution | $n_s$ | $n_t$ | mAP(%) |
|--------|----------|------------|-------|-------|--------|
| Q2L-R10T†(Liu et al., 2021a) | ResNet10T | $576 \times 576$ | - | - | 74.8 |
| HSQ-R10T(Ours) | ResNet10T | $576 \times 576$ | 0 | 1 | 71.5 |
| HSQ-Linear | ResNet10T | $576 \times 576$ | - | - | 75.7 |
| HSQ-R10T(Ours) | ResNet10T | $576 \times 576$ | 1 | 1 | 76.3 |
| HSQ-R10T(Ours) | ResNet10T | $576 \times 576$ | 4 | 1 | 76.1 |
| HSQ-R10T(Ours) | ResNet10T | $576 \times 576$ | 16 | 1 | **76.5** |

COCO to compare it with previous well-known works and state-of-the-art approaches. Results are shown in Table 2. We use ResNet101 (He et al., 2016) and ConvNeXt (Liu et al., 2022)(CvN) as the backbone and set input resolution to $576 \times 576$. Those backbones noted with -22k indicate that they are pre-trained on ImageNet-22k. Our HSQ model with CvN as the backbone achieves state-of-the-art performance at an mAP of 92.0%. Among all ResNet101-based approaches, our model outperforms all its counterparts. HSQ-R101 at the resolution of $576 \times 576$ achieves an mAP of 87.1%. Please note that for this model, we employ two successive Hybrid Sharing Layers of $d_o = 1024, 512$, MLP with one hidden layer of 128 neurons as gate and one of 64 as classifier.

Table 2: Performance (%) on MS-COCO. Bests are in bold. † indicates that it is not available in the original work and we implement it in this paper.

| Method | Backbone | Resolution | mAP | All | | Top3 | |
|--------|----------|------------|-----|-----|-----|------|-----|
| | | | | CF1 | OF1 | CF1 | OF1 |
| SRN (Zhu et al., 2017) | ResNet101 | $224 \times 224$ | 77.1 | 71.2 | 75.8 | 67.4 | 72.9 |
| ResNet-101 (He et al., 2016) | ResNet101 | $224 \times 224$ | 78.3 | 72.8 | 76.8 | 69.7 | 73.6 |
| CADM (Chen et al., 2019b) | ResNet101 | $448 \times 448$ | 82.3 | 77.0 | 79.6 | 73.5 | 76.0 |
| ML-GCN (Chen et al., 2019a) | ResNet101 | $448 \times 448$ | 83.0 | 78.0 | 80.3 | 74.2 | 76.3 |
| KSSNet (Liu et al., 2018) | ResNet101 | $448 \times 448$ | 83.7 | 77.2 | 81.5 | - | - |
| MS-CMA (You et al., 2020) | ResNet101 | $448 \times 448$ | 83.8 | 78.4 | 81.0 | 74.3 | 77.2 |
| MCAR (Gao and Zhou, 2021) | ResNet101 | $448 \times 448$ | 83.8 | 78.0 | 80.3 | 75.1 | 76.7 |
| SSGRL (Chen et al., 2019c) | ResNet101 | $576 \times 576$ | 83.8 | 76.8 | 79.7 | 72.7 | 76.2 |
| C-Trans (Lanchantin et al., 2021a) | ResNet101 | $576 \times 576$ | 85.1 | 79.9 | 81.7 | 76.0 | 77.6 |
| ADD-GCN (Ye et al., 2020) | ResNet101 | $576 \times 576$ | 85.2 | 80.1 | 82.0 | 75.8 | 77.9 |
| Q2L-R101 (Liu et al., 2021a) | ResNet101 | $448 \times 448$ | 84.9 | 79.3 | 81.5 | 73.3 | 75.4 |
| Q2L-R101 (Liu et al., 2021a) | ResNet101 | $576 \times 576$ | 86.5 | 81.0 | 82.8 | 76.5 | 78.3 |
| SST (Chen et al., 2022) | ResNet101 | $448 \times 448$ | 85.9 | 80.2 | 82.2 | 76.0 | 77.9 |
| ResNet101+TF (Xu et al., 2022) | ResNet101 | $576 \times 576$ | 85.9 | 80.3 | 82.4 | - | - |
| PSD+TF (Xu et al., 2022) | ResNet101 | $576 \times 576$ | 86.7 | 81.2 | 82.9 | - | - |
| SCO-DCNN (Zhang et al., 2023) | ResNet101 | $576 \times 576$ | 86.0 | 79.8 | 83.0 | - | - |
| HSQ-R101(Ours) | ResNet101 | $576 \times 576$ | **87.1** | **81.8** | **83.4** | **91.8** | **93.4** |
| ASL (Ridnik et al., 2021a) | TResNetL | $448 \times 448$ | 86.6 | 81.4 | 81.8 | 75.1 | 77.4 |
| TResNetL | TResNetL(22k) | $448 \times 448$ | 88.4 | - | - | - | - |
| Q2L-TResL (Liu et al., 2021a) | TResNetL | $448 \times 448$ | 87.3 | 81.6 | 83.1 | 77.0 | 78.5 |
| Q2L-TResL (Liu et al., 2021a) | TResNetL(22k) | $448 \times 448$ | 89.2 | 83.8 | 84.9 | 79.0 | 80.2 |
| MITr-l (Cheng et al., 2022) | MLTr-l(22k) | $384 \times 384$ | 88.5 | 83.3 | 84.9 | - | - |
| Swin-L (Liu et al., 2021b) | Swin-L(22k) | $384 \times 384$ | 89.6 | 84.8 | 86.1 | 80.0 | 81.1 |
| CvT-w24 (Wu et al., 2021) | CvT-w24(22k) | $384 \times 384$ | 90.5 | 85.4 | 86.6 | 80.3 | 81.3 |
| Q2L-SwinL (Liu et al., 2021a) | Swin-L(22k) | $384 \times 384$ | 90.5 | 85.4 | 86.4 | 80.5 | 81.2 |
| Q2L-CvT (Liu et al., 2021a) | CvT-w24(22k) | $384 \times 384$ | 91.3 | 85.9 | 86.8 | 80.8 | 81.6 |
| ML-Decoder†(Ridnik et al., 2023) | TResNet-XL(Open Image) | $640 \times 640$ | 91.2 | 76.8 | 76.9 | 90.8 | 92.0 |
| HSQ-CvN(Ours) | ConvNeXt(22k) | $576 \times 576$ | **92.0** | **86.6** | **87.5** | **94.0** | **95.2** |

### 4.3 PERFORMANCE ON THE VOC DATASET

PASCAL-VOC (Everingham et al., 2015) 2007 is also a well-acknowledged dataset for multi-label image classification. It comprises images of 20 classes and is split into train-val and test sets. We follow previous work to train on train-val set and validate on test set on 2007 version. The results of

Table 3: Performance (%) on VOC, in terms of per-label AP and mAP. Bests are in bold.

| Methods | aero | bike | bird | boat | bottle | bus | car | cat | chair | cow | mAP |
|---|---|---|---|---|---|---|---|---|---|---|---|
| CNN-RNN (Wang et al., 2016) | 96.7 | 83.1 | 94.2 | 92.8 | 61.2 | 82.1 | 89.1 | 94.2 | 64.2 | 83.6 | 84.0 |
| VGG+SVM (Simonyan and Zisserman, 2015) | 98.9 | 95.0 | 96.8 | 95.4 | 69.7 | 90.4 | 93.5 | 96.0 | 74.2 | 86.6 | 89.7 |
| Fev+Lv (Yang et al., 2016) | 97.9 | 97.0 | 96.6 | 94.6 | 73.6 | 93.9 | 96.5 | 95.5 | 73.7 | 90.3 | 90.6 |
| HCP (Wei et al., 2015) | 98.6 | 97.1 | 98.0 | 95.6 | 75.3 | 94.7 | 95.8 | 97.3 | 73.1 | 90.2 | 90.9 |
| RDAL (Wang et al., 2017) | 98.6 | 97.4 | 96.3 | 96.2 | 75.2 | 92.4 | 96.5 | 97.1 | 76.5 | 92.0 | 91.9 |
| RARL (Chen et al., 2018) | 98.6 | 97.1 | 97.1 | 95.5 | 75.6 | 92.8 | 96.8 | 97.3 | 78.3 | 92.2 | 92.0 |
| SSGRL(576) (Chen et al., 2019c) | 99.7 | 98.4 | 98.0 | 97.6 | 85.7 | 96.2 | 98.2 | 98.8 | 82.0 | 98.1 | 95.0 |
| MCAR (Gao and Zhou, 2021) | 99.7 | 99.0 | 98.5 | 98.2 | 85.4 | 96.9 | 97.4 | 98.9 | 83.7 | 95.5 | 94.8 |
| ASL(TResNetL) (Ridnik et al., 2021a) | **99.9** | 98.4 | 98.9 | 98.7 | 86.8 | 98.2 | 98.7 | 98.5 | 83.1 | 98.3 | 95.8 |
| ADD-GCN(576) (Ye et al., 2020) | 99.8 | 99.0 | 98.4 | 99.0 | 86.7 | 98.1 | 98.5 | 98.3 | 85.8 | 98.3 | 96.0 |
| Q2L-TResL (Liu et al., 2021a) | **99.9** | 98.9 | **99.0** | 98.4 | **87.7** | 98.6 | **98.8** | **99.1** | 84.5 | 98.3 | 96.1 |
| HSQ-CvN(22k) | **99.9** | **99.9** | 97.2 | **99.4** | 84.1 | **99.1** | 98.3 | **99.1** | **84.9** | **100.0** | **96.4** |

| Methods | table | dog | horse | mbike | person | plant | sheep | sofa | train | tv | mAP |
|---|---|---|---|---|---|---|---|---|---|---|---|
| CNN-RNN (Wang et al., 2016) | 70.0 | 92.4 | 91.7 | 84.2 | 93.7 | 59.8 | 93.2 | 75.3 | 99.7 | 78.6 | 84.0 |
| VGG+SVM (Simonyan and Zisserman, 2015) | 87.8 | 96.0 | 96.3 | 93.1 | 97.2 | 70.0 | 92.1 | 80.3 | 98.1 | 87.0 | 89.7 |
| Fev+Lv (Yang et al., 2016) | 82.8 | 95.4 | 97.7 | 95.9 | 98.6 | 77.6 | 88.7 | 78.0 | 98.3 | 89.0 | 90.6 |
| HCP (Wei et al., 2015) | 80.0 | 97.3 | 96.1 | 94.9 | 96.3 | 78.3 | 94.7 | 76.2 | 97.9 | 91.5 | 90.9 |
| RDAL (Wang et al., 2017) | 87.7 | 96.8 | 97.5 | 93.8 | 98.5 | 81.6 | 93.7 | 82.8 | 98.6 | 89.3 | 91.9 |
| RARL (Chen et al., 2018) | 87.6 | 96.9 | 96.7 | 93.6 | 98.5 | 81.6 | 93.1 | 83.2 | 98.5 | 89.3 | 92.0 |
| SSGRL(576) (Chen et al., 2019c) | 89.7 | 98.8 | 98.7 | 97.0 | 99.0 | 86.9 | 98.1 | 85.8 | 99.0 | 93.7 | 95.0 |
| MCAR (Gao and Zhou, 2021) | 88.8 | 99.1 | 98.2 | 95.1 | 99.1 | 84.8 | 97.1 | 87.8 | 98.3 | 94.8 | 94.8 |
| ASL(TResNetL) (Ridnik et al., 2021a) | 89.5 | 98.8 | 99.2 | 98.6 | **99.3** | 89.5 | 99.4 | 86.8 | 99.6 | 95.2 | 95.8 |
| ADD-GCN(576) (Ye et al., 2020) | 88.9 | 98.8 | 99.0 | 97.4 | 99.2 | 88.3 | 98.7 | 90.7 | 99.5 | 97.0 | 96.0 |
| Q2L-TResL (Liu et al., 2021a) | 89.2 | **99.2** | **99.2** | **99.2** | **99.3** | **90.2** | **98.8** | 88.3 | 99.5 | 95.5 | 96.1 |
| HSQ-CvN(22k) | **91.2** | **99.2** | **99.9** | **99.9** | 99.2 | 88.0 | **100.0** | **91.6** | **99.8** | **97.8** | **96.4** |

experiments on it are displayed in Table 3. Our model is compared against various established methods and state-of-the-art techniques. Notably, our proposed approach surpasses all its counterparts, achieving an impressive mAP score of 96.4%. The per-class average precision is also presented, with the SOTA performance bolded. Items that improve in comparison with previous works are underscored in the last two rows.

**Performance among Labels and Cross-label Comparison** Among the 20 available labels, our model exhibits superior performance in 15 of them. Compared to Q2L, another transformer-based model, our model improves 105 pairs of labels, 27 pairs more than Q2L achieved. To provide a visual comparison, we randomly select two labels with moderate performance (i.e., "table" and "train") and illustrate them in Figure 7. In this graph, each dot represents a specific approach, with dots in the upper-right corner indicating better performance. We further explore the performance difference between two pairs of labels with similar semantics, as depicted in Figure 3 and 4. Our method not only outperforms previous work but also exhibits a smaller absolute cross-label performance gap.

**Robust Performance across Multiple Image Scales.** In addition to previous experiments on MS-COCO, we perform extra experiments on different image scales to verify the performance of our model improves as image resolution decreases in Figure 6. We perform experiments on $576 \times 576$, $488 \times 448$ and $384 \times 384$. Results confirm that our model provides consistent considerable performance as image scale decreases.

## 4.4 VISUALIZATION RESULT

The proposed model incorporates several gates to harmonize outputs from experts. We verify that different tasks would rely on different experts on PASCAL VOC. Figure 5 depicts the weights of experts' outputs on all 20 tasks and the average load across tasks on a sampled data batch. The last row represents an expert's average load across all tasks. Weights are softmax-activated values of the gate networks' outputs, presented in log scale, along 33 different experts on the X-axis, 32 of which are shared, and the last one is respective task-specialized. A lighter block color indicates an expert with more weight in the final harmonized output. It is evident to note that all tasks focus on different experts. For instance, $E_{s,8}$, $E_{s,30}$, $E_{s,28}$ have the most significant impact on *chair*, *dog* and *horse* respectively. The weight distributions across experts exhibit variations among tasks, indicating that

distinct tasks rely on different sets of experts, each extracting distinctive representations, while even average loads on experts show that all experts are engaged during inference.

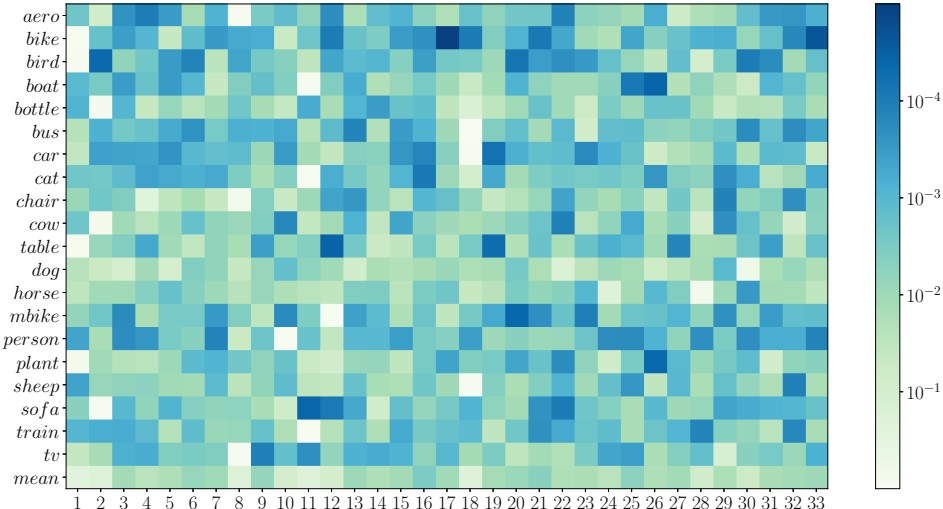

Figure 5: Experts load visualization on 20 labels of VOC2007. Each block indicates a weight for an expert on one task. The first 20 rows represent 20 labels from the VOC dataset, and the last row stands for the average load of experts. The x-axis denotes different experts, where experts 1-32 are shared among all tasks, while expert 33 is task-specific. The color represents weight in $\log$ space.

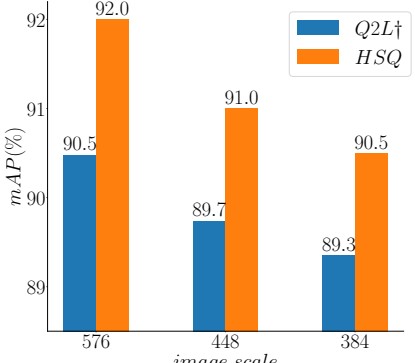

Figure 6: Performance comparison between Q2L (Liu et al., 2021a) and HSQ on MS-COCO with backbone as ConvNext (Liu et al., 2022)(22k)

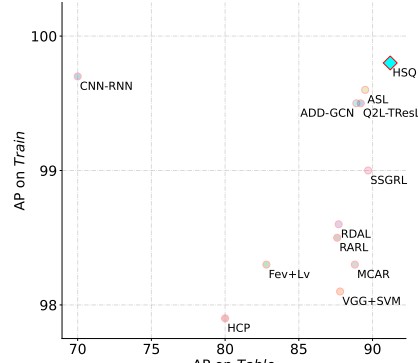

Figure 7: AP on VOC2007 (table and train). The upper-right points in the figure perform better. AP is in %

## 5  CONCLUSION

In this paper, regarding MLC as an MTL problem, we introduce HSQ, a transformer-based multi-label image classification model, which is constituted of a feature extraction backbone, query transformer, and Hybrid Sharing Layers that provide evident information sharing among tasks with shared and task-specialized experts leveraging inter- and intra-task information, respectively. Task-specialized experts are organized by respective gate networks, allowing each task to accept correlated information from shared experts independently. Shared experts accept input from all tasks, fusing all potentially useful information. Our model mitigates the negative transfer problem in MLC when formulating it as an MTL problem, where learning several labels jointly may hinder performance improvement. Our experiments demonstrate that HSQ provides a significant improvement on tested datasets. Furthermore, HSQ can simultaneously enhance per-label performance across multiple labels, mitigate performance gap among labels, and effectively handle semantic correlation and heterogeneity.

**Reproducibility Statement**    In this paper, we make efforts to provide detailed information to ensure the reproducibility and completeness of our work. Figure 1 illustrates the architecture of our model. Algorithm 1 and Figure 2 provide a clear overview and procedure for our crucial component, the Hybrid Sharing Layer. Section A.3 in the Appendix describes the hyper-parameters and devices we use, including optimizer, learning rate, *etc*. Section 4.2 and 4.3 describe details on how we prepare our dataset, including version, partitioning strategy, *etc*. The code will be available upon acceptance.

## ACKNOWLEDGMENTS

This work is partially supported by the National Natural Science Foundation of China (grant no. 62106101), and the Natural Science Foundation of Jiangsu Province (grant no. BK20210180). This work is also partially supported by the AI & AI for Science Project of Nanjing University.

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

## A APPENDIX

### A.1 METRICS DEFINITION

We here provide the definitions of metrics that are mentioned in our paper.

$$\text{Pre-Category Precision: } \text{CP} = \frac{1}{C} \sum_i \frac{N_i^c}{N_i^p}$$

$$\text{Pre-Category Recall: } \text{CR} = \frac{1}{C} \sum_i \frac{N_i^c}{N_i^g}$$

$$\text{Pre-Category F1 score: } \text{CF1} = \frac{2 \times \text{CP} \times \text{CR}}{\text{CP} + \text{CR}}$$

$$\text{Overall Precision: } \text{OP} = \frac{\sum_i N_i^c}{\sum_i N_i^p}$$

$$\text{Overall Recall: } \text{OR} = \frac{\sum_i N_i^c}{\sum_i N_i^g}$$

$$\text{Overall F1 score: } \text{OF1} = \frac{2 \times \text{OP} \times \text{OR}}{\text{OP} + \text{OR}}$$

where $C$ stands for the number of labels, $N_i^c$ refers to the number of samples that are correctly predicted for the $i$-th label, $N_i^p$ denotes the number of predicted samples for the $i$-th label, and $N_i^g$ means the number of ground truth samples for the $i$-th label.

### A.2 FLOPS AND #PARAMETERS DETAILS

| Model | Backbone | Resolution | FLOPs | #Parameters |
|---|---|---|---|---|
| Q2L-R10T†(Liu et al., 2021a) | ResNet10T | 576 | 8.4G | 14.8M |
| HSQ ($n_s = 1, n_t = 1$) | ResNet10T | 576 | 8.6G | 14.3M |
| HSQ ($n_s = 16, n_t = 1$) | ResNet10T | 576 | 8.7G | 15.5M |

### A.3 IMPLEMENTATION DETAILS

Unless otherwise stated, the following setting is valid for all experiments. We resize all input images from any dataset to $H_i \times W_i = 576 \times 576$. After a pre-trained backbone with timm, a convolutional layer projection would keep the embedding size $C = C_i = 2048$ as default.

Two layers of mixture-of-expert layers with output embedding dimension $d_o = 64, 16$ are implemented after a transformer with one encoder layer and two decoder layers, followed by a linear layer to make a final prediction.

We train the model for 100 epochs using the Adam optimizer, with weight decay of 1e-2, $(\beta 1, \beta 2)$ = (0.9, 0.9999), and a learning rate of 1e-4. All experiments are run on 4 Tesla V100-SXM2-32GB. The pre-training details of the experiments are provided in parentheses. We, by default, do not train our model on extra data, except if it is otherwise stated. Note that models with -22k indicate that they use a pre-trained backbone on ImageNet22k (Ridnik et al., 2021b). Additionally, experiments marked with † indicate that they have been replicated in this work.

