# OpenReview forum: "Hybrid Sharing for Multi-Label Image Classification"
_ICLR.cc/2024/Conference — ICLR 2024 poster_

### Official Review · Reviewer_7ihD · 2023-10-22

**Soundness:** 2 fair
**Presentation:** 4 excellent
**Contribution:** 2 fair
**Rating:** 6
**Confidence:** 5

**Summary:**

This paper presents a transformer-based approach with mixture-of-experts design for multi-label classification task.  The approach is design for better leveraging semantic correlation and heterogeneity among labels. The experiments on Pascal Voc and MSCOCO datasets show the effectiveness of the propose approach and get SoTA performance.

**Strengths:**

+ The paper offers valuable insights on multi-label classification, the solution based on MoE is a pretty novel view.

+ The approach achieves good results.

+ The paper is well-organized and have clear figures and demonstrations.

**Weaknesses:**

The main concern I have is in the experiment section.

1). The author missed some important experiments on MS-COCO.  The latest paper chosen for comparison is the ML-Decoder. However, the author didn't choose the same model backbone and the same image resolution to make a fair comparison. Besides, the model backbone pretraining details can also be an important factor in the model performance, so what's the difference between these methods?.

2). The ML-Decoder's code is available for years, the author should also fill the blanks in (CF1 OF1 CF1 OF1).

3). The ML-Decoder's paper and code is publicly available on arxiv at 2021, they didn't make any change to their best results even when they published to WACV 2023. This means all the results you compared with are before or in 2021, so why not choose some latest methods in 2022 and 2023 for comparison?

4). Please indicate the Flops and Parameters in the experiments and compared with the mentioned methods.

5). MSCOCO is a relatively small dataset with well annotated labels in the multi-label classification task. Why not using NUS-WIDE or OpenImages Dataset which are stronger benchmark and can also show the robustness of the model under the case with more labels and more data?

**Questions:**

Please answer the questions in [weaknesses].

---

> ### Author Response · Authors · 2023-11-20
> **Response to Reviewer 7ihD (1)**
>
> Thanks for your thorough evaluation and constructive feedback. We greatly value your insights. We have carefully revised our manuscript according to your suggestion. We will address all the concerns below.
>
> > W1: The author missed some important experiments on MS-COCO. The latest paper chosen for comparison is the ML-Decoder. However, the author didn't choose the same model backbone and the same image resolution to make a fair comparison. Besides, the model backbone pretraining details can also be an important factor in the model performance, so what's the difference between these methods?
> >
> > W2: The ML-Decoder's code is available for years, the author should also fill the blanks in (CF1 OF1 CF1 OF1).
> >
> > W3: The ML-Decoder's paper and code is publicly available on arxiv at 2021, they didn't make any change to their best results even when they published to WACV 2023. This means all the results you compared with are before or in 2021, so why not choose some latest methods in 2022 and 2023 for comparison?
>
> Thanks for your suggestions on our experiments. We are adding some contemporary works in image MLC for comparison and reconstruing Table 2. in our paper. We provide a comparison among models based on ResNet101 in the first part of Table 2. and all available items in the second part. We use model provided by timm library as our backbone. We do not use extra data unless it is otherwise stated. Please refer to the implement details section for default settings. For a fair comparison between identical backbones and settings, please refer to the ablation experiments and the first part of Table 2. Apart from these, Figure 6 also provides a comparison using the same backbone on 3 different image scales, showing our model provides a constant performance improvement.
>
> It is hard to reproduce a fair and reasonable comparison with ML-Decoder (ResNet101-based without extra data). We attribute this difficulty to the inherent nature of their research settings, which may not have been conducive to training from scratch. Despite this, we have re-run specific components of ML-Decoder on the COCO dataset. It's important to note that unstated variables have been retained in their default settings, as outlined in the official code. The figures are as follows.
>
> | Model      | Pretrain   | Backbone   | Resolution | mAP  | CF1      | OF1      | Top-3 CF1 | Top-3 OF1 |
> | ---------- | ---------- | ---------- | ---------- | ---- | -------- | -------- | --------- | --------- |
> | ML-Decoder | None       | ResNet101  | 576        | 66.1 | 57.8     | 61.77    | 75.5      | 80.2      |
> | ML-Decoder | COCO       | TResNetL   | 448        | 88.5 | 73.7     | 72.9     | 89.0      | 90.1      |
> | ML-Decoder | Open Image | TResNet-XL | 640        | 91.2 | 76.8 | 76.9 | 90.8  | 92.0      |
>
> We have already noticed that there is an ADDS [1] framework for multi-label classification reaching a very impressive performance. This framework employs NLP knowledge during their training. Given its multi-modal approach that incorporates substantial additional information from other domains, we consider it unsuitable and unfair for direct comparison. Therefore, we have chosen not to include it in the comparison. Since not all works provide top-3 metrics, we add a contemporary work with top-3 metrics despite it was not performed on the same image scale. Here are the reported figures on the COCO dataset compared with the proposed HSQ model.
>
> | Model           | Backbone  | Resolution | mAP      | CF1      | OF1      | Top-3 CF1 | Top-3 OF1 |
> | --------------- | --------- | ---------- | -------- | -------- | -------- | --------- | --------- |
> | HSQ             | ResNet101 | 576        | **87.1** | **81.8** | **83.4** | **91.8**  | **93.4**  |
> | SST[2]          | ResNet101 | 448        | 85.9     | 80.2     | 82.2     | 76.0      | 77.9      |
> | ResNet101+TF[3] | ResNet101 | 576        | 85.9     | 80.3     | 82.4     | -         | -         |
> | PSD+TF[3]       | ResNet101 | 576        | 86.7     | 81.2     | 82.9     | -         | -         |
> | SCO-DCNN[4]     | ResNet101 | 576        | 86.0     | 79.8     | 83.0     | -         | -         |

---

> ### Author Response · Authors · 2023-11-20
> **Response to Reviewer 7ihD (2)**
>
> > W4: Please indicate the Flops and Parameters in the experiments and compared with the mentioned methods.
>
> | Model                | Backbone  | Resolution | FLOPS | #Parameters |
> | -------------------- | --------- | ---------- | ----- | ----------- |
> | Q2L†                 | ResNet10t | 576        | 8.4G  | 14.8M       |
> | HSQ ($n_s=1,n_t=1$)  | ResNet10t | 576        | 8.6G  | 14.3M       |
> | HSQ ($n_s=16,n_t=1$) | ResNet10t | 576        | 8.7G  | 15.5M       |
>
> We provide a comparison between the Q2L baseline and the proposed HSQ. We noticed that HSQ and Q2L shares very close size of FLOPS and #Parameters and increasing the size of shared experts won't increase the FLOPS and #Parameters dramatically.  Nevertheless, those aforementioned contemporary up-to-date works did not report FLOPs and #Parameters in their papers or provide official code.
>
> > W5: MSCOCO is a relatively small dataset with well annotated labels in the multi-label classification task. Why not using NUS-WIDE or Open Images Dataset which are stronger benchmark and can also show the robustness of the model under the case with more labels and more data?
>
> Thank you for your valuable suggestion.  In our review of the literature, we've noted that while the NUS-WIDE dataset has been referenced in some papers, the COCO dataset and VOC dataset are more widely adopted for comparisons in visual multi-label classification studies, as seen in comparable works. The COCO dataset with 80 classes aligns closely with the NUS-WIDE dataset, which contains 81 classes, suggesting a similar level of task (label) complexity.
>
> We believe that conducting experiments on the COCO and VOC datasets would allow for comparable assessments with previous studies. This approach ensures that our model's performance is evaluated across datasets of varying sizes, ranging from small to large in the context of the number of labels/tasks. Nevertheless, we thank you for your kind suggestion on these two datasets. We are also actively considering them in the future work on this topic.
>
> Thank you for your time, and we sincerely appreciate your valuable and constructive suggestions. We hope that our answer will effectively address any concerns you may have had regarding our paper.
>
> **Reference**
>
> > [1] He, S., Guo, T., Dai, T., Qiao, R., Shu, X., Ren, B., & Xia, S. T. (2023, June). Open-vocabulary multi-label classification via multi-modal knowledge transfer. In *Proceedings of the AAAI Conference on Artificial Intelligence* (Vol. 37, No. 1, pp. 808-816).
> >
> > [2] Chen, Z.-M., Cui, Q., Zhao, B., Song, R., Zhang, X., & Yoshie, O. (2022). SST: Spatial and Semantic Transformers for Multi-Label Image Recognition. IEEE Transactions on Image Processing, 31, 2570–2583. https://doi.org/10.1109/TIP.2022.3148867
> >
> > [3] Xu, J., Huang, S., Zhou, F., Huangfu, L., Zeng, D., & Liu, B. (2022). Boosting Multi-Label Image Classification with Complementary Parallel Self-Distillation. 2, 1495–1501. https://doi.org/10.24963/ijcai.2022/208
> >
> > [4] Zhang, J., Ren, J., Zhang, Q., Liu, J., & Jiang, X. (2023). Spatial Context-Aware Object-Attentional Network for Multi-Label Image Classification. IEEE Transactions on Image Processing, 32, 3000–3012. https://doi.org/10.1109/TIP.2023.3266161

---

> > ### Comment · Reviewer_7ihD · 2023-11-22
> >
> > Thanks for your reply, and they mostly addressed my concerns.

---

### Official Review · Reviewer_rQYR · 2023-10-30

**Soundness:** 3 good
**Presentation:** 2 fair
**Contribution:** 2 fair
**Rating:** 6
**Confidence:** 5

**Summary:**

In this work, the authors regard the multi-label image recognition task as a multi-task problem. To address the "label heterogeneity", the authors propose a transformer-based model, Hybrid Sharing Query (HSQ), that introduces the mixture-of-experts architecture to multilabel classification, which leverages label correlations while mitigating heterogeneity effectively. As presented in their experiment results, the proposed framework achieves state-of-the-art performance, which demonstrates the effectiveness of the proposed method.

**Strengths:**

1. It is novel that presenting a "mixture of experts" to the multi-label image recognition task, which is ignored in previous MLR works.
2. The design of the proposed framework is technically clear, and the experiment results demonstrate its effectiveness.

**Weaknesses:**

1. As the core motivation, the authors should provide a detailed and comprehensive discussion about "label heterogeneity" in MLR.
2. In the hybrid sharing layer, what is the difference between the task-specialized experts group and the semantic features in other MLR works? And what is the shared experts group? These are crucial for understanding this work.
3. Could you provide some visualizations of the impact of the shared experts group? Considering the claim "shared experts help to extract correlations between tasks, encourage positive correlation sharing and suppress negative transfer"

**Questions:**

Please see the above weaknesses.

---

> ### Author Response · Authors · 2023-11-20
> **Response to Reviewer rQYR**
>
> Thank you for your thoughtful and detailed feedback. Below are our detailed responses to your concerns.
>
> > W1: As the core motivation, the authors should provide a detailed and comprehensive discussion about "label heterogeneity" in MLR.
>
> In this paper, we identify the prediction on every possible label as a task in multi-task learning (MTL). Previous  research on MTL has proven that learning one task can hinder progess on another. Despite the MTL model typically improving on average performance, performance on some particular tasks may deteriorate [1, 2]. This can be viewed as a negative transfer problem,  wherein the model learns something from a task that obstacles its learning. Some works also term this as the see-saw phenomenon, as improving on one task usually decreases the performance on another. As this paper considers MLC as a special case in MTL, performance on some labels may deteriorate due to the joint learning paradigm on all labels. We believe there is underlying correlation and heterogeneity among all labels (tasks) features in MLC, where correlation is something that should be shared across labels, and heterogeneity may hinder joint performance improvement. The proposed mechanism of a designed architecture will help to adaptively fuse these features, suppressing the heterogeneity among labels while allowing correlation to be shared across labels.
>
> > W2: In the hybrid sharing layer, what is the difference between the task-specialized experts group and the semantic features in other MLR works? And what is the shared experts group? These are crucial for understanding this work.
>
> As previously mentioned and detailed in our paper, by framing  the MLC problem as a special case of MTL, we believe learning on one label (task) may hinder another. Features exhibit variability among labels, and not all of them should be universally shared. For example, consider a scenario with three labels. Learning on label A may enhance the performance of label B and vice versa, thus joint learning on label A and B is beneficial to both of them. We want to encourage these parts of features (label correlation) to be shared between label A and B through the utilization of shared experts. Conversely, for label C, learning on both A and B may hinder learning on C (due to label heterogeneity). To address this, we employ task-specialized experts to exploit per-task features. The gate of this task will fuse these features adaptively. This approach allows us to navigate the challenges posed by label heterogeneity and optimize learning on each specific task.
>
> > W3: Could you provide some visualizations of the impact of the shared experts group? Considering the claim "shared experts help to extract correlations between tasks, encourage positive correlation sharing and suppress negative transfer"
>
> Please refer to the ablation study in our paper (Table 1). One may notice a prominent performance decline when $n_s=0$ compared with the baseline without expert and $n_s=4$ run. We believe it proves the significance of shared experts. When n_s is set to 0, there is no shared expert presented in the model, meaning that cross-label (task) information sharing among the experts is entirely blocked, proving the sharing experts contribute a lot to the performance. We are adding this part of analysis to our paper.
>
> Please also kindly refer to Figure 5 in our paper. Figure 5 depicts the loads of experts on every label (task). We do not see any recognizable pattern among all labels from Aero to TV, meaning every task relies on a different mixture of experts, learning its unique weight distribution on experts. The last row indicates the per-expert mean load on every label. The mean experts’ load is evenly distributed, showing no load balancing problem is present.
>
> Thank you for your time, and we sincerely appreciate your valuable and constructive suggestions. We hope that our answer will effectively address any concerns you may have had regarding our paper.
>
> **Reference**
>
> > [1] Wu, S., Zhang, H. R., & Ré, C. (2019, September). Understanding and Improving Information Transfer in Multi-Task Learning. In International Conference on Learning Representations.
> >
> > [2] Liu, S., Liang, Y., & Gitter, A. (2019, July). Loss-balanced task weighting to reduce negative transfer in multi-task learning. In Proceedings of the AAAI conference on artificial intelligence (Vol. 33, No. 01, pp. 9977-9978).

---

### Official Review · Reviewer_2H28 · 2023-10-31

**Soundness:** 3 good
**Presentation:** 3 good
**Contribution:** 2 fair
**Rating:** 6
**Confidence:** 4

**Summary:**

The paper addresses the issue of heterogeneity in the multi-label learning process. It transforms multi-label tasks into multi-task learning and innovatively incorporates the MoE model into visual classification tasks. Drawing inspiration from models like MMoE and PLE, the authors employed multiple experts and gating theories to learn various labels in their experiments. To prevent negative transfer during the model learning process, they also employed a shared expert approach to learn label correlations. Finally, the authors conducted experiments on the VOC and MS-COCO datasets and achieved outstanding performance.

**Strengths:**

1. Innovatively introducing the MoE model from the NLP field into the domain of computer vision, the paper achieved outstanding results. The use of gating and expert effectively enhanced the model's capacity for relationship modeling in multi-label tasks.
2. The paper's algorithmic description is concise and clear, with pseudo-code illustrating the core model workflow.
3. The experiments are comprehensive, including model comparisons, ablation studies, and visualizations.

**Weaknesses:**

1. In the ablation experiments section, there is a lack of explanation for the decrease in experimental performance, particularly why the performance declines when n_t is 1 and n_s is 0, and why is there not a more in-depth exploration of the impact of the quantities of n_s and n_t on the results?
2. There is a lack of visualization of how different experts in the model process images.
3. Contribution 1 and Contribution 2 appear quite similar. The experiments on heterogeneity are not sufficiently intuitive, why is it solely demonstrated through experiments rather than being theoretically proven?
4. The paper uses extensive textual descriptions in the methodology section; using formulas would provide a more concise representation.
5. Although the paper incorporates the MoE model, it has relatively few innovative aspects of its own.

**Questions:**

1. What is the significance of Figure 6? The reduction in image resolution implies a decrease in information, which may lead to a decline in the model's performance. How does this relate to robustness?
2. What is the difference between hybrid experts and attention mechanisms? Is the model's good performance due to the introduction of a large number of parameters? Does MoE in the paper improve training speed? It is recommended to conduct an efficiency experiment to verify this question.
3. MoE models generally encounter the issue of load balancing. Is there a possibility that certain experts consistently dominate during the model training process? It is recommended to conduct an experiment to verify this question.

---

> ### Author Response · Authors · 2023-11-20
> **Response to Reviewer 2H28 (1)**
>
> We sincerely appreciate your thorough review and valuable feedback.  We have carefully revised our manuscript according to your suggestion. We will address all the concerns point by point.
>
> > W1: In the ablation experiments section, there is a lack of explanation for the decrease in experimental performance, particularly why the performance declines when n_t is 1 and n_s is 0, and why is there not a more in-depth exploration of the impact of the quantities of n_s and n_t on the results?
>
> Thanks for your suggestion. We have conducted several additional experiments to answer this question.
>
> | Model      | Backbone  | Resolution | $n_s$ | $n_t$ | mAP  |
> | ---------- | --------- | ---------- | ----- | ----- | ---- |
> | Q2L        | ResNet10T | 576        | -     | -     | 74.8 |
> | HSQ        | ResNet10T | 576        | 0     | 1     | 71.5 |
> | HSQ-Linear | ResNet10T | 576        | -     | -     | 75.7 |
> | HSQ        | ResNet10T | 576        | 1     | 1     | 76.4 |
> | HSQ        | ResNet10T | 576        | 2     | 1     | 76.3 |
> | HSQ        | ResNet10T | 576        | 4     | 1     | 76.1 |
> | HSQ        | ResNet10T | 576        | 6     | 1     | 76.0 |
> | HSQ        | ResNet10T | 576        | 16    | 1     | 76.5 |
>
> We maintain consistent configurations across all HSQ runs. Specifically, in the HSQ-Linear configuration, we only replace the hybrid sharing layers with linear layers, ensuring that the `in_features` and `out_features` match the hybrid sharing layers. Notably, there is a discernible performance decline in Table 1 when $n_t=1$,$n_s=0$. Our analysis attributes this decline to the crucial nature of information exchange among labels, even if some information may adversely affect performance. When $n_s$ is set to 0, the model excludes any shared experts. Consequently, per-label (task) features are not shared among experts, and each label only utilizes features from its respective task-specialized experts, resulting in a complete blockade of information sharing. This comprehensive cutoff of information sharing among labels emerges as the key factor contributing to the observed decline in performance.
>
> To validate that merely adding shared information without distinguishing task-specialized details would not yield similar performance, we conducted an experiment with HSQ-Linear. Here, we simply replace the hybrid sharing layers with fully connected linear layers while maintaining the depth of the model, as well as keeping `in_features` and `out_features` identical to those in HSQ. HSQ-Linear forgoes the use of a mixture-of-experts mechanism and instead employs fully-connected layers where the MoE would typically be implemented. It shares all information across all labels and does not distinguish task-specialized information. We observe a persistent performance gap between HSQ-Linear and our proposed HSQs, underscoring the contribution of hybrid sharing layers to performance improvement.
>
> The primary focus of this paper is on the interaction among all labels (tasks), considering both correlation and heterogeneity among labels. Our approach involves task-specialized experts designed to extract intra-label features. While increasing the number of task-specific experts ($n_t$) would quickly introduce a significant number of parameters, it offers limited assistance in substantiating the main theme of our paper.
>
> We believe that augmenting the number of experts, both in terms of task-specific ($n_t$) and shared ($n_s$) experts, may enhance overall performance, especially when employing a more robust backbone or dealing with a complex dataset. However, it's essential to note that simply increasing the number of task-specific experts may not be the optimal strategy, as it introduces a substantial parameter overhead without necessarily contributing substantially to the core focus of our research.
>
> We are adding this part of the analysis to our paper. Thanks again for your kind suggestion.
>
> > W2: There is a lack of visualization of how different experts in the model process images.
>
> Please refer to Figure 5 for a detailed analysis of this problem. In Figure 5, we present how every task relies on experts’ output. Each row in the figure represents the weight of a task on the various experts. Notably, we did not observe any discernible pattern across all available labels (tasks). This implies that, for each label (task), there exists a distinct and unique distribution of weights assigned to the experts.

---

> ### Author Response · Authors · 2023-11-20
> **Response to Reviewer 2H28 (2)**
>
> > W3: Contribution 1 and Contribution 2 appear quite similar. The experiments on heterogeneity are not sufficiently intuitive, why is it solely demonstrated through experiments rather than being theoretically proven?
>
> The concept of label correlation and heterogeneity are manifestations of positive transfer and negative transfer in multi-task learning, and the phenomena of positive transfer and negative transfer have been widely acknowledged in the context of multi-task learning. Previous works like [1, 2, 3] gave adequate empirical observation and theoretical discussion on this topic in various domains like NLP, recommending systems, etc. Recent work has attributed negative transfer to conflict between the directions and magnitudes of task gradients [3].
>
> > W4: The paper uses extensive textual descriptions in the methodology section; using formulas would provide a more concise representation.
>
> Thanks for your kind suggestions, we had some formulas in our old version. We found it took up too much space and removed it. We think it would be a cliché to use formulas to describe the backbone and transformer parts. We are adding a formula to help readers to comprehend how the shared experts work.
>
> > W5: Although the paper incorporates the MoE model, it has relatively few innovative aspects of its own.
>
> This paper aims to provide a new view on the multi-label classification task by regarding MLC as a multi-task learning problem, which is rarely explored in prior works. This aspect enables us to re-understand the bottleneck of performance improvement in MLC. As it has been argued and studied in MTL, learning a label (task) may hinder another, even though the overall performance is improved. By incorporating the MoE mechanism, we show that MoE is a promising and underestimated mechanism in CV. Our model presents a dedicated mechanism to fuse task-specialized and shared experts to fit this image MLC task. Experiments have proven that our model achieves both overall performance improvement and more pre-task performance improvement as well as mitigating performance gap among labels.
>
> > Q1: What is the significance of Figure 6? The reduction in image resolution implies a decrease in information, which may lead to a decline in the model's performance. How does this relate to robustness?
>
> Figure 6 aims to demonstrate that the improvement of our model is robust to image scale change. Our model shows consistent performance improvement across all tested image scales.
>
> > Q2: What is the difference between hybrid experts and attention mechanisms? Is the model's good performance due to the introduction of a large number of parameters? Does MoE in the paper improve training speed? It is recommended to conduct an efficiency experiment to verify this question
>
> The attention mechanism and hybrid experts / MoE mechanism are different. The dense fusion strategy of MoE is somewhat similar to the attention mechanism. MoE mechanism aims to incorporate several experts and fuse the outputs. In the NLP context, every expert is designed to be responsible for a lexical category, improving performance by widening rather than deepening the model. The fusion strategy of MoE can be categorized into gating in dense MoE and routing in sparse MoE, where dense MoE activates all experts during inference while sparse activates a subset of experts. Hybrid experts consist of task-specialized experts and shared experts. Please refer to Figure 2. for a detailed illustration. In general, the MoE mechanism aims to widen the model to capture a wider range of features and adaptively fuse them together. As the model gets wider, it commonly does not provide any acceleration on training speed. Previous works [4, 5] have argued that employing experts:
>
> 1) facilitates the specialization of individual experts on smaller facets of a broader problem
>
> 2) employs soft partitions of the data
>
> 3) it enables the formation of splits along hyperplanes oriented arbitrarily within the input space. These attributes substantiate the model's capacity to represent nonstationary or piecewise continuous data within a sophisticated regression framework and to discern nonlinearities within a classification problem.
>
> | Model                | Backbone  | Resolution | FLOPS | #Parameters |
> | -------------------- | --------- | ---------- | ----- | ----------- |
> | Q2L†                 | ResNet10t | 576        | 8.4G  | 14.8M       |
> | HSQ ($n_s=1,n_t=1$)  | ResNet10t | 576        | 8.6G  | 14.3M       |
> | HSQ ($n_s=16,n_t=1$) | ResNet10t | 576        | 8.7G  | 15.5M       |
>
> We provide a comparison between the Q2L baseline and the proposed HSQ. We noticed that HSQ and Q2L shares very close size of FLOPS and #Parameters and increasing the size of shared experts won't increase the FLOPS and #Parameters dramatically.
>
> We believe that good performance does not come with a simple concatenation of a great number of parameters. The architecture of our model contributes to the final improvement.

---

> ### Author Response · Authors · 2023-11-20
> **Response to Reviewer 2H28 (3)**
>
> > Q3: MoE models generally encounter the issue of load balancing. Is there a possibility that certain experts consistently dominate during the model training process? It is recommended to conduct an experiment to verify this question.
>
> MoE models do suffer from load balancing problems when they have bad router/gate networks. We have compared loads of experts on each task in our paper. Please refer to Figure 5 in the original version. Each row of Figure 5 represents the weights of every expert. Expert's loads on the label Areo~Tv do not show any recognizable pattern, proving the load of experts varies across all tasks. The average expert load (shown on the last row) is smooth, showing that all experts are equally utilized.
>
> Thank you for your time, and we sincerely appreciate your valuable and constructive suggestions. We hope that our answers will effectively address any concerns you may have had regarding our paper.
>
> **Reference**
>
> >  [1] Wu, S., Zhang, H. R., & Ré, C. (2019, September). Understanding and Improving Information Transfer in Multi-Task Learning. In *International Conference on Learning Representations*. 2020
> >
> > [2] Progressive Layered Extraction (PLE): A Novel Multi-Task Learning (MTL) Model for Personalized Recommendations | Proceedings of the 14th ACM Conference on Recommender Systems. (n.d.).
> >
> > [3] Mueller, D., Andrews, N., & Dredze, M. (2022). Do Text-to-Text Multi-Task Learners Suffer from Task Conflict? In Y. Goldberg, Z. Kozareva, & Y. Zhang (Eds.), Findings of the Association for Computational Linguistics: EMNLP 2022 (pp. 2843–2858). Association for Computational Linguistics.
> >
> > [4] S. E. Yuksel, J. N. Wilson and P. D. Gader, "Twenty Years of Mixture of Experts," in IEEE Transactions on Neural Networks and Learning Systems, vol. 23, no. 8, pp. 1177-1193, Aug. 2012, doi: 10.1109/TNNLS.2012.2200299.
> >
> > [5] Z. Chen, Y. Deng, Y. Wu, Q. Gu, and Y. Li, ‘Towards Understanding the Mixture-of-Experts Layer in Deep Learning’, Advances in Neural Information Processing Systems, vol. 35, pp. 23049–23062, Dec. 2022.

---

> > ### Comment · Reviewer_2H28 · 2023-11-22
> > **Thanks for the replies!**
> >
> > Thank you for the response and the additional experiments, the introduction of MoE does indeed seem to enhance the model's performance and addresses most of my concerns. I've raised my score to 6.

---

### Author Response · Authors · 2023-11-20
**To all reviewers**

We express our gratitude to the reviewer for their thoughtful assessment and detailed feedback. After carefully considering all reviews, we are implementing the following revisions to our paper:

1. We are adding an analysis of the notable performance decline in our ablation study.
2. We are adding extra runs to our ablation study in Table 1 and extra contemporary works in Table 2.
3. We are moving the implementation details to the appendix to comply with the conference page limitation.
4. We are adding a Table to describe FLOPS and #Parameters to the Appendix

The aforementioned revisions have been made.

Revisions are marked in red color.

---

### Meta-Review · Area_Chair_AXAb · 2023-12-07

**Metareview:**

This paper regards the multi-label image recognition task as a multi-task problem. To address the "label heterogeneity", the authors propose a transformer-based model, Hybrid Sharing Query (HSQ), that introduces the mixture-of-experts architecture to multilabel classification, which leverages label correlations while mitigating heterogeneity effectively. The approach is design for better leveraging semantic correlation and heterogeneity among labels. The experiments on Pascal Voc and MSCOCO datasets show the effectiveness of the propose approach and get SoTA performance.

Overall, all three reviewers give positive rating. The reviewers agree that the solution based on MoE is pretty sound and the experiments are comprehensive. Based on the reviews, I lean towards acceptance.

**Justification For Why Not Higher Score:**

There are some concerns regarding to the dataset size in experiments (by reviewer 7ihD), the presentation is somehow unclear (lack of visualization, by reviewer 2H28 and rQYR).

**Justification For Why Not Lower Score:**

Please refer to metareviewer section.

---

### Decision · Program_Chairs · 2024-01-16

Accept (poster)